# Friedel-Crafts-Type Acylation and Amidation Reactions in Strong Brønsted Acid: Taming Superelectrophiles [note 1]

**DOI:** 10.3390/molecules27185984

**Published:** 2022-09-14

**Authors:** Akinari Sumita, Tomohiko Ohwada

**Affiliations:** Graduate School of Pharmaceutical Sciences, The University of Tokyo, 7-3-1 Hongo, Bunkyo-ku, Tokyo 113-0033, Japan

**Keywords:** superelectrophile, isocyanate cation, acylium cation, amidation, acylation, aminocarboxylic acid, phosphoric acid esters, charge-charge repulsion

## Abstract

In this review, we discuss Friedel-Crafts-type aromatic amidation and acylation reactions, not exhaustively, but mainly based on our research results. The electrophilic species involved are isocyanate cation and acylium cation, respectively, and both have a common ^+^C=O structure, which can be generated from carboxylic acid functionalities in a strong Brønsted acid. Carbamates substituted with methyl salicylate can be easily ionized to the isocyanate cation upon (di)protonation of the salicylate. Carboxylic acids can be used directly as a source of acylium cations. However, aminocarboxylic acids are inert in acidic media because two positively charged sites, ammonium and acylium cation, will be generated, resulting in energetically unfavorable charge-charge repulsion. Nevertheless, the aromatic acylation of aminocarboxylic acids can be achieved by using tailored phosphoric acid esters as Lewis bases to abrogate the charge-charge repulsion. Both examples tame the superelectrophilic character.

## 1. Introduction: Acylium Ions

The Friedel-Crafts reactions are reactions of cationic carbon electrophilic species with an aromatic compound, enabling carbon substituents to be introduced onto the aromatic ring. [1] Indeed, the aromatic acylation reaction [2] is one of the most useful reactions in organic synthesis, particularly in medicinal chemistry, because many medicines contain an aromatic system that participates in hydrophobic interactions. In aromatic acylation, the cationic electrophilic species that reacts with the aromatic compound is an acylium ion. This is usually generated from carboxylic acid derivatives such as carboxylic anhydride and acid chloride under strongly acidic conditions, such as in the presence of aluminum trichloride and sulfuric acid. The structures of some acylium ions have been determined by means of NMR spectroscopy [3,4] and X-ray crystallography [5,6].

Effenberger et al. examined the reactivity of isolated and purified acylium ion salts with alkylbenzenes to give **3** (Figure 1) [7]. They showed that the acylium ion salt (**2**) is in equilibrium with the trifluoromethanesulfonate form (**1**) in the reaction solvent (1,2-dichloroethane) (Figure 1). When the electron density of the carboxylic acid is high (**1**, R = MeO, Figure 1) the equilibrium is biased towards the acylium ion (**2**) and the electrophilicity is lower. On the other hand, when the electron density of the carboxylic acid is low (**1**, R = NO_2_, Figure 1), the contribution of the acylium ion (**2**) in the equilibrium is small, but the electrophilicity is greater.

Another recent study [8] concluded that acylium ion salts are the active species in the aromatic acylation reaction, even though the acylium ion precursor (a complex of acid chloride and Lewis acid) is observed as the main component in the solvent.

Nevertheless, the electrophilicity of acylium ions is not high and thus they react efficiently only with electron-rich benzenes, but not with non-activated aromatic compounds such as benzene and halobenzenes. Therefore, in order to improve the electrophilicity of acylium ions, research has been focused on superacids, [9] which have higher acidity than 100% sulfuric acid.

It is well known that the reactivity of acylium ions increases with increasing acidity of the reaction solution. Ohwada and Shudo et al. also reported a relationship between the acidity of the reaction medium and the efficiency of the aromatic acylation reaction (Figure 2) [10]. In the reaction of an isolated acylium ion salt (**4**) with benzene, they found that the yield of the target aromatic ketone (**5**) was low in TFA (acidity function *−H*_0_ = 2.7), but increased as the acidity of the reaction medium was increased (Figure 2).

The acylium dication has been proposed to play a role in the high efficiency of this reaction (Figure 3). That is, the monocation (**4**) is converted to a dication (**6**) or protosolvated form (**7**) by further protonation or hydrogen bonding, thereby increasing its electrophilicity. 

Such multivalent cations with enhanced electrophilicity have been called superelectrophiles, and various kinds of superelectrophile species have been reported [11,12].

Thus, in the aromatic acylation reaction, higher acidity of the reaction medium generally accelerates the reaction. However, there are still some cases in which the desired acylation does not proceed even when the acidity is increased. Two examples are discussed below.

Klumpp et al. reported the reactivity of cinnamic acid (**8**) in very strong acids (Figure 4) [13]. They found that if the aromatic moiety of cinnamic acid (**8**) has an electron density higher than that of halobenzene, the aromatic acylation reaction does not proceed to give the aromatic ketone (**10**), but benzene reacts at the β-position of the olefin and the resultant saturated carboxylic acid undergoes an intramolecular aromatic acylation reaction to give the cyclized product (**9**). This is because the acylium ion is conjugated to the styrene moiety, so that the cationic carbon is switched from the carbonyl carbon (**C_C=O_**) to the benzyl (β-)carbon (**C**), as in **12** (Figure 5) [14]. Therefore, the desired aromatic ketone formation does not proceed because the aromatic ring reacts with the benzyl carbon atom first.

Another example is aminocarboxylic acid. Olah et al. examined the structure of amino acids in very strong acids, and they found that an amino acid (**13**) with a sufficiently long carbon chain between the amino and carboxyl groups could form a dication (**14**), that is, protonation of the amino nitrogen atom and ionization to the acylium ion both occurred (Figure 6a) [15]. On the other hand, when an α-amino acid (**15**) such as valine was used, a dication was formed, but protonation took place at the amino group and the carboxyl group, and ionization to afford the acylium ion (**17**) did not occur due to charge-charge repulsion (Figure 6b) [16,17,18,19,20]. While protonation of the amino nitrogen atom and the carbonyl oxygen atom can occur, ionization to form the acylium ion (**17**) in superacid depends strongly on the energy requirement for C–O bond cleavage during acylium ion formation, due to the repulsion between the positive charges (Figure 6b). Therefore, the aromatic acylation reaction of aminocarboxylic acids has been little studied so far. 

Aromatic ketones, the products of aromatic acylation reactions, are components of various bioactive substances and pharmaceuticals, and are also useful as synthetic intermediates [21]. Therefore, if we can overcome the problem of charge-charge repulsion and control the reaction, we can expect to achieve concise syntheses of a variety of useful compounds.

Here, we review our efforts to solve these problems, focusing especially on activation of the carboxylic acid functionality. We discuss aromatic amidation and acylation reactions. The electrophilic species involved are isocyanate cation and acylium cation, respectively, and both have a common ^+^C=O structure, which can be generated from carboxylic acid functionalities in a strong Brønsted acid. Carbamates substituted with methyl salicylate can be easily ionized to the isocyanate cation upon (di)protonation of the salicylate. Carboxylic acids can be used directly as a source of acylium cations. However, aminocarboxylic acids are inert in acidic media because two positively charged sites, ammonium and acylium cation, will be generated, resulting in energetically unfavorable charge-charge repulsion. The activation of the leaving group by using methyl salicylate was not valid to the aromatic acylation of aminocarboxylic acids. Nevertheless, the aromatic acylation of aminocarboxylic acids can be achieved by using tailored phosphoric acid esters as Lewis bases to abrogate the charge-charge repulsion. Both examples tame the superelectrophilic character. 

## 2. Aromatic Amidation

### 2.1. Utility of Methyl Salicylate as a Leaving Group in Generation of Electrophiles 

In the aromatic acylation reaction, carboxylic acids are converted to acid chlorides or anhydrides, and then the acyl group is introduced into aromatic compounds via acylium ion formation under acidic conditions. However, acid chlorides and acid anhydrides have limited chemical stability. In contrast, Olah et al. employed chemically stable methyl esters in the aromatic acylation reaction [22]. In their method, methyl esters are activated under strongly acidic conditions to produce acylium ions. This is a practical approach from the viewpoint of synthetic chemistry. However, because it requires heating in strong acid, which may result in the decomposition of other functional groups, there is still a need for further improvement of the methodology.

Salicylic acid has long been known as a good leaving group [23]. Olah et al. focused on the intramolecular hydrogen bond formation of salicylic acid and examined protonation of the phenolic oxygen atom (as in **18**, Figure 7). They found that acetylsalicylic acid was diprotonated to give the dication **19** at −40 °C (Figure 7) [24]. However on warming to 0 °C, the dication **19** was transformed to monocation **20** and acyl cation **21** (Figure 7). This suggests that salicylic acid can form an intramolecular hydrogen bond and serve as a good leaving group to form the acylium ion **21**. This aromatic acylation reaction has the potential to be a versatile synthetic method.

### 2.2. Aromatic Amidation Reaction Using Methyl Salicylate as a Leaving Group

Our group has reported a method for generating isocyanate cations by using methyl salicylate as a leaving group (Figure 8) [25,26]. The carbamate functional group consisting of isocyanate and methyl salicylate is chemically stable [27] and poorly reactive, but under strongly acidic conditions, the isocyanate cation (**23** or **27**) is quickly generated at room temperature by the cleavage of methyl salicylate (Figure 8) [28,29,30]. The formed isocyanate cation reacts intramolecularly with the aromatic moiety to afford the aromatic lactam **24** [31,32]. Furthermore, when **25** and **26** react intermolecularly, the aromatic amide (**28**) can be generated through the isocyanate intermediate (**27**) via a similar process (Figure 8).

However, the efficiency of the reaction was drastically reduced for chemical species with a cationic charge in the vicinity. This suggests that the reaction between carbamate and the aromatic compound does not proceed via the A_ac_2 mechanism, in which the aromatic ring reacts with the carbamate functional group and generates a tetrahedral intermediate, but rather via the A_ac_1 mechanism, in which the electrophilic species (isocyanate cation) produced by the elimination of methyl salicylate from the carbamate reacts with the aromatic ring. In the case of dicarbamate (**29**), one methyl salicylate was not cleaved, and the mono-amide (**31**) was formed in 32% yield, probably because the cleavage of salicylate from the intermediate **32** to form the acylium cation **33** was slow (Figure 9). This is likely due to charge-charge repulsion between the protonated amide and the forming isocyanate cation **33** [26].

Thus, we next synthesized carbamate (**35**) (Figure 10), whose cleavage ability was improved by introducing electron-withdrawing *o*, *p*-bis(methyl salicylate). We found that the reaction with an aromatic compound (**36**) in a strong acid gave diamide **37** in 66% yield (Figure 10) [26]. These experimental results suggest that it is possible to control the reactivity of carbamates by regulating their cleavage capacity.

### 2.3. Aromatic Acylation Reaction Using Methyl Salicylate as a Leaving Group

If the good cleavage ability of methyl salicylate is applicable to esters, the formation of acylium ions can be expected (Figure 11). Therefore, we acetylated the phenolic hydroxyl group of methyl salicylate. Indeed, the ester **38** reacted rapidly with the aromatic compound in a strong acid to afford the desired aromatic ketone (**39**) in 83% yield at 0 °C (Figure 11) [33]. The rate of elimination of methyl salicylate from the ester **38** is greater than that from the corresponding carbamate (**40**). At 0 °C, the carbamate (**40**) was stable for at least 90 min at 0 °C and the starting carbamate was recovered in 89% yield, while at 20 °C, the corresponding amide **41** was formed in 85% yield after 60 min (Figure 11). This result indicates that the chemoselectivity of this reaction can be kinetically controlled by adjusting the temperature. Mechanistically charge-charge repulsion of the intermediate dications **B** and **D** was weakened by separation of the cationic centers (Figure 11).

### 2.4. Difference in Cleavage Ability

The difference in the reactivity of carbamates and esters having methyl salicylate as a leaving group (Figure 11) can be explained on the basis of the DFT calculations (Figure 12). We believe the real active cationic species are dictations **B** and **D** in which the two cationic centers are separated and distal, but in these calculations, the equilibrating monocations **A** and **C** were calculated (Figure 11).

The formation of acylium ion (**43**) from the protonated ester (**42**) requires cleavage of the C–O bond between the carbonyl carbon and the phenolic hydroxyl group of methyl salicylate through the TS (**44**). In this case, the intramolecular hydrogen bond of methyl salicylate reduces the activation energy required for C–O bond cleavage, resulting in rapid acylium ion formation (∆*G^‡^*_298K_ = 13.1 [kcal/mol], ∆*H^‡^* = 13.7 [kcal/mol]). On the other hand, the carbamate (**45**) forms Y-type conjugation [27] around the carbonyl carbon atom, which increases the activation energy of C–O bond cleavage between the carbonyl carbon and the phenolic hydroxyl group of methyl salicylate through the TS (**47**) (∆*G^‡^*_298K_ = 16.1 [kcal/mol], ∆*H^‡^* = 17.1 [kcal/mol]). Therefore, isocyanate cation (**46**) formation from the carbamate (**40**) is slower than acylium ion formation from the ester [33].

The DFT calculations also showed that in strong acids, carbamate is most stable in an 8-membered ring structure (**40**) with intramolecular hydrogen bonding between the methyl ester group carbonyl oxygen atom of methyl salicylate and the carbamate. However, in the case of the ester, the structure with intramolecular hydrogen bonding between the phenolic oxygen atom and the methyl ester carbonyl oxygen atom of methyl salicylate (**42**) appears to be more stable than the 8-membered ring structure (**38**). Therefore, in the case of the ester group, the acylium ion is rapidly formed from the intramolecular hydrogen-bonded state, but in the case of the carbamate group, the total activation energy is increased because it takes extra energy to convert the 8-membered ring state to the structure **45**.

### 2.5. Tandem Reactions

The cleavage capacity was found to affect the rate of generation of the electrophilic species (isocyanate cation and acylium cation). The reaction of the electrophilic species with aromatic compounds proceeds rapidly, suggesting that the electrophilic reaction can be controlled by varying the rate of generation of the electrophilic species, i.e., by differences in cleavage capacity. This idea can be extended to the tandem reactions (Figure 13) of indole [34], indene [35,36], dihydroindene [37], indanone [38], fluorene [39,40,41], carbazole [42,43,44], diphenylmethane-triphenylmethane [45,46], naphthoquinone [47,48,49,50,51], and so on. While the focus is on skeleton formation [52], the ability to link sub-skeletons in this reaction is attractive.

When the ester (**48**) and carbamate (**49**), containing o-methyl salicylate as a leaving group react in a strong acid at 0 °C, *o*-methyl salicylate is selectively removed from the ester group and reacts with the aromatic ring present in the carbamate (**49**) in an intermolecular reaction to give the aromatic ketone (**50**) (Figure 13). Subsequently, another carbamate (**51**) bearing an aromatic ring is added to the reaction solution, and when the temperature is raised to room temperature (20 °C), *o*-methyl salicylate is cleaved from the carbamate (**50**) and reacts intermolecularly with the aromatic part B of the carbamate (**51**) to give an aromatic amide (**52**) (Figure 13). After a sufficient reaction time (15 hr), isocyanate cation is also formed from the carbamate (**52**) incorporating *p*-methyl salicylate to give an aromatic amide via intramolecular cyclization (**53**) (Figure 13). These three electrophilic reactions can be conducted sequentially in one pot by adjusting the leaving group, reaction temperature and reaction time [33].

### 2.6. Limitations of the Present Reaction System

In the reaction using *o*-methyl salicylate as the leaving group, aromatic ketone (**55**) is rapidly synthesized from the active ester (**54**), even at low temperature (Figure 14).

However, when methyl salicylate is used as a leaving group, the efficiency of cleavage from anthranilic acid, which has an amino group in the ortho position, is not high (Figure 14). Even when the ester (**56**), a condensation product of anthranilic acid with *o*-methyl salicylate, was reacted with benzene in TfOH, the desired aromatic ketone (**57**) was not formed, and only the starting material was recovered (Figure 14). The reason for this is thought to be inhibition of acylium ion formation by the protonated amino nitrogen atom, that is charge-charge repulsion [16].

## 3. Aromatic Acylation Reaction with Phosphoric Acid Esters and Strong Bronsted Acid

### 3.1. Phosphoric Acid Esters

Phosphoric acid esters have high oxygen affinity and are commonly utilized by enzymes such as glutamine synthetase [52,53,54,55] and aminoacyl-tRNA synthetase [56,57,58,59] to activate carboxylic acids in living organisms. Acyl phosphates [60,61,62,63,64,65,66] can react with a variety of nucleophiles, are highly reactive with carboxylic acids, and can mediate efficient functional group conversion reactions, as exemplified by synthetic reagents such as polyphosphate [67], Eaton reagents [68,69], DPPA reagents [70], and BOP reagents [71].

Therefore, by utilizing phosphoric acid esters with high oxygen affinity, aromatic acylation reactions with various carboxylic acids can be expected, even where the reactivity is reduced due to charge-charge repulsion. However, phosphoric acid esters themselves are chemically stable [72,73,74] and require electrophilic activation.

Salicylic acid or methyl salicylate has been used for electrophilic activation of phosphoric acid esters (Figure 15), and Bender et al. attributed the high hydrolysis reaction rate of salicylic acid-linked phosphoric acid monoesters (**54**) to the presence of the intramolecular hydrogen bond between the carboxyl and phenolic hydroxyl groups (Figure 15a) [75]. Brown et al. also reported that the reaction of phosphoric acid diesters (**57**) with methyl salicylate-bound methanol under strongly acidic conditions proceeded rapidly (Figure 15b) [76]. They attributed the high reaction rate to the coordination of the yttrium cation to methyl salicylate and the non-bridged oxygen atom of the phosphoric acid ester (**58**), which enhanced the cleavage capacity of methyl salicylate (Figure 15b). As described above, salicylic acid and methyl salicylate are electrophilically active chemical species that enhance the cleavage capacity by forming noncovalent bonds to protons and cationic species, making the phosphoric acid ester electrophilic. Kirby et al. also reported the high reactivity of a phosphoric acid diester (**61**) with two salicylic acids attached, proposing intramolecular nucleophilic reaction of the carbonyl oxygen atom of salicylic acid, which is not a leaving group, to phosphorus (Figure 15c) [77]. The intermediate (**62**) formed by the intramolecular nucleophilic reaction accelerates the cleavage of salicylic acid. Thus, salicylic acid and methyl salicylate not only serve as good leaving groups, but may also enhance the cleavage capacity of other ester linkers [78,79].

These results suggest that by using salicylic esters, which are chemically stable and easy to synthesize, as phosphate linkers, various electrophilic activation pathways can be generated in the phosphoric acid ester. This enables the reaction with the carboxylic acid to proceed rapidly to afford the acylium ion, followed by aromatic acylation.

### 3.2. Aromatic Acylation of Aminocarboxylic Acids

The reaction of benzoic acid (**65**) with benzene in trifluoromethanesulfonic acid (TfOH) in the presence of phosphoric acid triester (**67**), a triester of *o*-methyl salicylate, at room temperature for 20 min gave the desired aromatic ketone (**68**) in a yield of 92% (Figure 16, Table 1, Entry 1) [80]. In a control experiment in the absence of **67**, the desired aromatic ketone (**68**) was not obtained after reaction for 20 min. When the reaction time was extended to 24 h, the aromatic ketone (**68**) was produced even in the absence of **67**, but in only 48% yield (Table 1, Entries 2 and 3). Thus, it was suggested that phosphoric acid triester (**67**) promotes the reaction of benzoic acid (**65**) with benzene, and can be regarded as an organocatalyst. When a much weaker acid, trifluoroacetic acid (TFA) was used, the desired aromatic ketone (**68**) was not formed, but ester **70** was produced (Table 1, Entry 4). Thus, the aromatic acylation reaction requires a strong acid.

Anthranilic acid (**66**), which bears an amino group at the ortho position, is also available as a substrate. While anthranilic acid (**66**) in TfOH produced the ketone **69** in less than 7% yield even after 24 hr (Figure 16, Table 1, Entries 6 and 7), the reaction proceeded in 88% yield in the presence of **67** (Table 1, Entry 5) [80]. When the acidity was lowered (acidity function *H*_0_ = −11.8) by mixing TfOH with the weaker acid TFA, the yield of aromatic ketone (**69**) decreased to 10 % and an ester (**71**) was formed in 38% yield through the reaction of methyl salicylate with anthranilic acid (Table 1, Entry 8). When the acidity was further reduced (trifluoroacetic acid; *H*_0_ = −2.7), the aromatic ketone (**69**) was not formed and **67** was recovered in 99% yield.

These reactions are noteworthy, because the acylium cation is difficult to form from anthranilic acid, as shown in Figure 16. The aromatic amine is basic enough to be pronated in the acid media, and the subsequent ionization of the carboxylic acid functionality to the acylium cation is blocked due to charge-charge repulsion (Figure 17) [16,17,18,19,20]. Thus, the promoting effect of phosphoric acid triester **67** plays a key role.

### 3.3. Characterization of Phosphoric Acid Esters of Methyl Salicylate

The reactivities of various phosphoric acid esters were examined (Figure 18) [80]. A combination of *o*-methyl salicylate (**72**) and methyl 4-hydroxybenzoate (**73**) changed the promoting activity in the aromatic acylation reaction of benzoic acid (**65**) with benzene in TfOH to give the ketone **68**. As the amount of *para*-isomer (**73**) moieties increased in the phosphoric acid triesters in the order **67** → **74** → **75** → **76**, the chemical yield of the ketone (**68**) decreased; that is, the promoting effect decreased (Figure 18). The tri-*p*-isomers **68** showed no promoting effect on the aromatic acylation. 

The yield of aromatic ketone (**68**) was greatly increased when the phosphoric acid ester contained two (**74**) or three (**67**) methyl salicylate linkers rather than a single methyl salicylate (**75**). In other words, the ortho ester group of methyl salicylate, which is a potent leaving group, functions to promote P–O bond formation with an external carboxylic acid, **65**, leading to ester exchange (Figure 19).

### 3.4. Structure of Phosphoric Acid Ester in Strong Acid

A strong acid is needed for the reaction to proceed (Table 1, Entries 1 and 4). Therefore, we investigated the structure of the phosphoric acid ester (**67**) under strongly acidic conditions [80].

The ^1^H NMR spectrum of **67** in trifluoromethanesulfonic acid showed a peak at low field (about 15 ppm), that was assigned to a proton forming an intramolecular hydrogen bond between the phenolic oxygen atom of one methyl salicylate and the carbonyl oxygen atom of the ortho ester group [25]. On the other hand, the NMR signal of the proton bound to the phosphate non-bridged oxygen atom is difficult to observe due to fast proton exchange with the solvent (CF_3_SO_3_–H) [25].

In the ^31^P NMR spectrum, the peak of **67** was observed at about −19 ppm in CDCl_3_ and in TFA, whereas in TfOH the signal shifted significantly toward high field at about −78 ppm. Olah et al. [81] reported that a significant change in ^31^P NMR peak values is usually not observed upon protonation of phosphoric acid esters at unbridged oxygen atoms. In addition, the ^31^P NMR peak values of the positional isomers **74** (−19 ppm) and **75** (−19 ppm) in CDCl_3_ are similar to those in TfOH (**74**: −21 ppm, **75**: −24 ppm). Thus, we consider that the significant shift in the ^31^P NMR peak values of **67** in TfOH is attributable to a structural change (Figure 20).

Possible structures of the phosphoric acid triester (**67**) include tetrahedral (**78**), trigonal bipyramidal (**79**), and octahedral (**80**) forms (Figure 20). The variety of structures is due to the different possible interactions of the carbonyl oxygen atom(s) of the *ortho* ester group with the phosphorus atom [82,83].

The ^31^P NMR chemical shifts were calculated for the diprotonated form, which is protonated at the non-bridged oxygen atom and in one methyl salicylate, of each of the structures shown in Figure 20. The calculated ^31^P chemical shift of the trigonal bipyramidal structure (**79**) was closest to the experimental value (^31^P NMR = −78 ppm). Therefore, there are two possible reaction pathways leading to P–O bonding of carboxylic acid **65** (Figure 21): **path a** involves dissociation of methyl salicylate first from dication **79**, followed by addition of the carboxylic acid (**65**), while **path b** involves addition of the carboxylic acid (**65**) to dication **79**, followed by dissociation of methyl salicylate. Experimentally, the presence and intermediacy of **77** were confirmed (see the following section), but distinguishing the two pathways is not easy. The DFT calculations suggest that **path b** is marginally more favorable than **path a**.

### 3.5. Experimental Evidence for the P–O Bond Formation of Carboxylic Acid

In order to detect the intermediate **77**, kinetic analysis using ^31^P NMR was carried out during the reaction of phosphoric acid triester (**67**) and benzoic acid (**65**). Figure 22 shows the ^31^P NMR spectral changes of a mixture of **67** and **65** in TfOH, obtained at 2-min intervals. Note that 85% H_3_PO_4_ is used as a standard. In the presence of benzoic acid (**65**), a new peak appears at around −6 ppm along with a decrease in the peak of **67** (^31^P NMR = −78 ppm). This new peak was assigned to acyl phosphate (**77**) formed by the reaction of **67** and benzoic acid (**65**) (Figure 22).

The structure of **77** was confirmed by high-resolution mass spectroscopy. Furthermore, after structural optimization by the DFT method, the NMR chemical shift value of the monoprotonated acyl phosphate **77** was calculated to be −6 ppm. This supports the view that the new peak at around −6 ppm is due to acyl phosphate (**77**) formed by the reaction of benzoic acid (**65**) and phosphoric acid triester (**67**).

In order to check the involvement of **path a** (Figure 20), we also tried to identify the dicationic species **81** (Figure 21) in a solution of **67** in TfOH. As the peak of **67** (^31^P NMR = −78 ppm) decreased (Figure 20), a new peak emerged at around −19 ppm (Figure 22). This peak was identified as a new phosphate cationic species [84] (**81**), which is formed from **67** by the removal of one methyl salicylate **72**. When the ^31^P NMR chemical shift value was calculated for the DFT-optimized phosphorus cation species (**81**), the predicted value was −19 ppm, in good agreement with the experimental value. Furthermore, treatment of the reaction solution with an excess amount of methanol under basic conditions at −78 °C yielded equal amounts of methyl salicylate (**72**) and phosphoric acid methyl ester (**82**) in 46% yield (Figure 23). Compound **82** is formed by the addition of methanol to the cation **81**. These experimental results indicate that the phosphoric acid triester (**67**) releases one methyl salicylate in TfOH through **79** and is converted to the cationic species (**81**), in which the phosphorus atom is stabilized by the carbonyl oxygen atom of the ortho ester group of the neighboring methyl salicylate (Figure 23). The results suggest that the phosphorus cation **81** is rather stable, but reacts with an excess of methanol to give the phosphoric acid triester (**82**). This latter process is similar to **path a** (Figure 21), in which the cation **81** reacts with the carboxylic acid (**65**). This indicates that **path a** is experimentally plausible. 

### 3.6. Reactivity of Acyl Phosphate

To understand the high reactivity of acyl phosphate itself, we synthesized acyl phosphates (**83** and **85**) using phenol without any substituent on the aromatic ring as a phosphate linker [85] and examined the reaction with benzene in a strong acid (Figure 24). [80] The desired aromatic ketone (**68**) was obtained in 82% yield (Figure 24). This experimental result is similar to that obtained with the phosphoric acid triester (**67**), suggesting that acyl phosphates themselves (**83**) have high reactivity and react quickly with aromatic compounds under strongly acidic conditions to form aromatic ketones. Under similar reaction conditions, the acyl phosphate (**85**) containing an anthranilic acid moiety reacted with benzene in TfOH, but the chemical yield of the ketone (**69**) was only moderate (40% yield). This is in sharp contrast to the results for the putative intermediate **77** (63% yield, Figure 22) and the phosphoric acid triester **67** (92% yield, Figure 18). 

### 3.7. Computed Reaction Profile of Acyl Phosphate Containing an Anthranilic Acid Moiety

The reason for the high reactivity of acyl phosphates is probably their high cleavage capacity, in spite of charge-charge repulsion in the resultant cation in the case of acyl phosphates of anthranilic acid (Figure 17).

The reaction profiles of carboxylic acid (**66**), ester (**86**) using methyl salicylate as the leaving group [86], and acyl phosphate (**87**) to form the respective acylium ion were examined by means of DFT calculations (Figure 25). In the case of anthranilic acid, diprotonation can occur at both amino nitrogen and carbonyl oxygen. However, dehydration does not occur to generate the acylium cation due to charge-charge repulsion (Figure 17).

One of the possible starting structures is intramolecularly hydrogen-bonded **66** (Figure 25). Another candidate is an ester of methyl salicylate (**86**), in which an intramolecular hydrogen bond can be formed within methyl salicylate. In this case, a counter anion was placed in the vicinity of the proton in order to avoid charge-charge repulsion in the product acylium cation. We compared the reaction profiles of **66** and **86** with that of the acyl phosphate **87**. The activation energy was low in the case of acyl phosphate **87**, probably due to stabilization by the hydrogen-bonding network, in particular in **TS-C** structure (Figure 25) [80].

The high reactivity of acyl phosphates, and thus the high cleavage capacity of phosphate diesters, can be explained in terms of resonance effects within the phosphate diesters.

### 3.8. Substrate Generality

The generality of substrate carboxylic acids was next examined (Figure 26) [80]. In this reaction, the desired aromatic ketone can be rapidly synthesized from various carboxylic acids. In particular, carboxylic acids bearing a basic amine group can serve as substrates (Figure 26). The reaction proceeds efficiently for amino acids having an unprotected aromatic or aliphatic amino groups (**91**–**100** and **102**). Other carboxylic acids such as conjugated carboxylic acids (**101**, **103**–**104**) and other electrophilic functionalized benzenic acid derivatives (**106**–**109**) also reacted well (Figure 26).

### 3.9. Application to 2,3-Benzodiazepine Skeleton Construction

A concise synthesis of the 2,3-benzodiazepine skeleton can be achieved by using the aromatic acylation reaction with phosphoric acid triester (**67**).

The 2,3-benzodiazepine skeleton (Figure 27) is an important scaffold in medicinal chemistry, especially for drugs related to the central nervous system (CNS), because of its pharmacological activity towards AMPA receptors [87]. There have been reports of side effects such as drug-dependence [88], but this is a more serious problem with the structurally isomeric 1,4-benzodiazepine derivatives [89]. Therefore, although some efficient construction methods for the 2,3-benzodiazepine skeleton have been reported recently, new methods remain of interest.

Recently reported methods for the construction of the 2,3-benzodiazepine skeleton are illustrated in Figure 28. Chan et al. constructed the skeleton by employing the Wacker reaction to produce a diketone followed by cyclization with hydrazine (Figure 28a) [90]. Zhu et al. applied the C–H activation reaction of an aromatic hydrazone compound with rhodium (III) to form the 2,3-benzodiazepine skeleton (Figure 28b) [91]. Okuma et al. used a benzyne precursor and 1,3-diketone to form a diketone, which was cyclized with hydrazine (Figure 28c) [92].

Further, the acylation reaction directly produces diketones or their derivatives, benzopyrylium salts, from aromatic rings with a ketone group at the β-position, and this is expected to provide a concise synthetic route from commercial reagents (Figure 28d) [93,94]. However, synthetic methods using the aromatic acylation reaction generally produce benzopyrylium salts quickly, and the synthesis of the 2,3-benzodiazepine skeleton is largely dependent on the reaction of the benzopyrylium salt with hydrazine. In particular, the reaction of benzopyrylium salts with hydrazine having an aliphatic alkyl chain attached to the C1 carbon precedes the nucleophilic reaction at the C1 carbon to give the benzoisoquinolium salt (Figure 29) [95]. The conversion of benzoisoquinolium salts to 2,3-benzodiazepine skeletons involves ring-opening reactions of the heterocyclic quinolium skeleton [96], which is generally inefficient.

The aromatic acylation reaction proceeds quickly at room temperature in the presence of phosphoric acid triester (**67**), so in this case, the acylation is expected to be complete before the benzopyrylium salt is formed (see Figure 29) [97].

When we tried to synthesize diketone (**112**) from an aromatic compound (**110**) and 4-aminobenzoic acid (**111**) using **67** in TfOH, we found that the diketone (**112**) gradually decomposed during purification (Figure 30, method (1)). The reason for this is thought to be that the diketone (**112**) and pyrylium ion (**113**) are in equilibrium, and both compounds are unstable [98]. Therefore, we examined the synthesis of nerizopam (**114**) by reacting the crude product of the aromatic acylation reaction with hydrazine in ethanol solution after aqueous work-up, and obtained the desired compound (**114**) in moderate yield (48%) (Figure 30, method (2)). Then, since the diketone (**112**) and pyrylium ion (**113**) seemed to gradually decompose during work-up, we developed a method in which hydrazine is added to the aromatic acylation reaction vessel together with base (Figure 30, method (3)).

Method 3 (Figure 30) gave nerizopam **114** in 73% yield. Next, the scope and limitations of this method for synthesizing 2,3-benzodiazepine derivatives were examined (Figure 31).

## 4. Summary

The isocyanate cation and the acylium cation can be generated by the elimination of methyl salicylate from the corresponding carbamate and ester compounds (Figure 11). Charge-charge repulsion was weakened in the dication intermediates (**B** and **D**) by tuning hydrogen bonding (Figure 11). If the reaction sites are multiple, charge-charge repulsion determined the reaction order, which enabled the tandem reactions (Figure 13). On the other hand, the acylium cation is difficult to generate from anthranilic acid in a strong acid, as shown in Figure 14, because the aromatic amine nitrogen atom is sufficiently basic to be protonated in acidic media. Thus, subsequent ionization of the carboxylic acid functionality to the acylium cation is blocked due to charge-charge repulsion, even though the leaving group ability was increased by using *o*-methyl salicylates (Figure 14 and Figure 17) [16,17,18,19,20]. However, phosphoric acid triester **67** works as a Lewis base to neutralize the cationic character, enabling these reactions to be conducted efficiently. Both examples of aromatic amidation and acylation, discussed here indicated taming superelectrophilicity is crucial to activate electrophilicity and to reduce destabilization due to charge-charge repulsion at the same time. 

## Data Availability

Not applicable.

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
