# Peer review of "Friedel-Crafts-Type Acylation and Amidation Reactions in Strong Brønsted Acid: Taming Superelectrophiles [Author-notes fn1-molecules-27-05984]"

_molecules, 2022, doi:10.3390/molecules27185984_

Round 1

Reviewer 1 Report

This review by Ohwada & Sumita discusses amidation and acylation reactions of activated arenes with isocyanate cation and acylium cations generated from carboxylic acid derivatives of salicylates or with aid of  phosphoric acid triester of salicylates in TfOH reported by themselves. It may appeal to readers in this field, however, the scope is rather narrow. I suggest a publication as a personal account rather than a common review if there is this type in Molecules. In addition, there are a couple of typos and formats to be corrected before publication, for example, but not limited to,

1) Please use italic style in o/m/p-methyl and the likes throughout the manuscript.

2) Check structure 62 in Scheme 15, a C=O missed there.

3) Please use bold style for all the number for compounds throughout the manuscript, in particular, in the titles of tables.

4) Page 11, Table 2-1 should be Table 1.

5) Please reword “This reaction requires strong acidity for the reaction to proceed”.

6) The formats of references between refs. 35 and 45 are in total mess.

Author Response

Thank you for the comments.

We revised the main text in accordance to the reviewer's comments 1)-6)

We changed the title of this paper . In the Abstract we added the following sentences: In this review, we discuss Friedel-Crafts-type aromatic amidation and acylation reactions, not exhaustively, but mainly based on our research results. 

Reviewer 2 Report

The title needs to be changed. Otherwise the paper is publishable in Molecules in its current form.

Author Response

We agree with the change of the title. We changed the title of this manuscript to Friedel-Crafts-type Acylation and Amidation Reactions in Strong Brønsted Acid: Taming Superelectrophiles.